



**Water adsorption and hygroscopic growth of six anemophilous pollen species: the effect of temperature**

Mingjin Tang,[1,5,6,*] Wenjun Gu,[1,5] Qingxin Ma,[2,5,6,*] Yong Jie Li,[3] Cheng Zhong,[2,5] Sheng Li,[1,5] Xin Yin,[1,5] Ru-Jin Huang,[4] Hong He,[2,5,6] Xinming Wang[1,5,6]

[1] State Key Laboratory of Organic Geochemistry and Guangdong Key Laboratory of Environmental Protection and Resources Utilization, Guangzhou Institute of Geochemistry, Chinese Academy of Sciences, Guangzhou 510640, China

[2] State Key Joint Laboratory of Environment Simulation and Pollution Control, Research Center for Eco-Environmental Sciences, Chinese Academy of Sciences, Beijing 100085, China

[3] Department of Civil and Environmental Engineering, Faculty of Science and Technology, University of Macau, Avenida da Universidade, Taipa, Macau, China

[4] Key Laboratory of Aerosol Chemistry and Physics, State Key Laboratory of Loess and Quaternary Geology, Institute of Earth and Environment, Chinese Academy of Sciences, Xi'an 710061, China

[5] University of Chinese Academy of Sciences, Beijing 100049, China

[6] Center for Excellence in Regional Atmospheric Environment, Institute of Urban Environment, Chinese Academy of Sciences, Xiamen 361021, China

* Correspondence: Mingjin Tang (mingjintang@gig.ac.cn), Qingxin Ma (qxma@rcees.ac.cn)



## Abstract

Hygroscopicity largely affects environmental and climatic impacts of pollen grains, one
important type of primary biological aerosol particles in the troposphere. However, our knowledge
in pollen hygroscopicity is rather limited, and especially the effect of temperature has rarely been
explored before. In this work three different techniques, including a vapor sorption analyzer,
diffusion reflectance infrared Fourier transform spectroscopy (DRIFTS) and transmission Fourier
transform infrared spectroscopy (transmission FTIR) were employed to characterize six
anemophilous pollen species and to investigate their hygroscopic properties as a function of
relative humidity (RH, up to 95%) and temperature (5 or 15, 25 and 37 $^{o}$C). Substantial mass
increase due to water uptake was observed for all the six pollen species, and at 25 $^{o}$C the relative
mass increase at 90% RH, when compared to that at <1% RH, ranged from ~30 to ~50%, varying
with pollen species. The modified $\kappa$-Köhler theory can well approximate the mass hygroscopic
growth of all the six pollen species, and the single hygroscopicity parameter ($\kappa$) was determined
to be in the range of 0.034±0.001 to 0.061±0.007 at 25 $^{o}$C. In-situ DRIFTS measurements
suggested that water adsorption by pollen species was mainly contributed by OH groups of organic
compounds they contained. Good correlations were indeed found between hygroscopicity of
pollen grains and the amount of OH groups, as determined using transmission FTIR. Increase in
temperature would in general lead to decrease in hygroscopicity, except for pecan pollen. For
example, $\kappa$ values decreased from 0.073±0.006 at 5 $^{o}$C to 0.061±0.007 at 25 $^{o}$C and to 0.057±0.004
at 37 $^{o}$C for populus tremuloides pollen, and decreased from 0.060±0.001 at 15 $^{o}$C to 0.054±0.001
at 25 $^{o}$C to 0.050±0.002 at 37 $^{o}$C for paper mulberry pollen.



## 1 Introduction

Primary biological aerosol particles (PBAPs), an important type of aerosol particles in the troposphere, are directly emitted from the biosphere and include pollen, fungal spores, bacteria, viruses, algae, and so on (Després et al., 2012; Fröhlich-Nowoisky et al., 2016). Emission and abundance of PBAPs are quite uncertain, and annual emission fluxes are estimated to be in the range of <10 to ~1000 Tg for total PBAPs and 47-84 Tg for pollen (Després et al., 2012). Pollen, and PBAPs in general, are of great concerns due to their various impacts on the Earth system (Sun and Ariya, 2006; Ariya et al., 2009; Georgakopoulos et al., 2009; Morris et al., 2011; Morris et al., 2014; Fröhlich-Nowoisky et al., 2016). For example, they can be allergenic, infectious or even toxic, affecting the health of human and other species in the ecological systems over different scales (Douwes et al., 2003; Reinmuth-Selzle et al., 2017; Shiraiwa et al., 2017). The geographical dispersion of anemophilous plants largely relies on pollen dispersal, which in turn depends on the emission, transport and deposition of pollen grains; therefore, pollen plays a key role in the evolution of many ecosystems (Womack et al., 2010; Fröhlich-Nowoisky et al., 2016). In addition, PBAPs can serve as giant cloud condensation nuclei (CCN) and ice nucleating particles (INPs), significantly impacting the formation and properties of clouds and thus radiative balance and precipitation (Möhler et al., 2007; Ariya et al., 2009; Pratt et al., 2009; Pope, 2010; Pummer et al., 2012; Gute and Abbatt, 2018). It has also been proposed that PBAPs may have significant impacts on chemical composition of aerosol particles via heterogeneous and multiphase chemistry (Deguillaume et al., 2008; Estillore et al., 2016; Reinmuth-Selzle et al., 2017; Shiraiwa et al., 2017).

Hygroscopicity is one of the most important physicochemical properties of pollen (as well as aerosol particles in general). Hygroscopicity largely impacts the transport and deposition of pollen grains (Sofiev et al., 2006), therefore affecting their lifetimes, abundance and



spatiotemporal distribution. In addition, hygroscopicity is closely linked to the ability of aerosol
particles to serve as CCN and INPs (Petters and Kreidenweis, 2007; Kreidenweis and Asa-Awuku,
2014; Tang et al., 2016). Several previous studies have measured the hygroscopicity and CCN
activities of pollen (Diehl et al., 2001; Pope, 2010; Griffiths et al., 2012; Lin et al., 2015; Steiner
et al., 2015; Prisle et al., 2018) and other PBAPs such as bacteria (Pasanen et al., 1991; Reponen
et al., 1996; Franc and DeMott, 1998; Ko et al., 2000; Lee et al., 2002; Bauer et al., 2003). For
example, water uptake of eleven pollen species was studied using an analytical balance (Diehl et
al., 2001), and the mass of pollen was found to be increased by 3-16% at 73% RH and by ~100-
300% at 95% RH, compared to that at 0% RH. An electrodynamic balance was employed to
investigate the hygroscopic growth of eight types of pollen (Pope, 2010; Griffiths et al., 2012), and
it was found that their hygroscopic growth can be approximated by the modified $\kappa$-Köhler theory,
with single hygroscopicity parameters being around 0.1 (depending on the assumed pollen density).

Previous measurements were mostly carried out at or close to room temperature, and the

effects of temperature on hygroscopic properties of pollen and other types of PBAPs are yet to be
elucidated. To our knowledge, only one previous study (Bunderson and Levetin, 2015) explored
the effect of temperature (4, 15 and 20 $^{o}$C) on the water uptake by Juniperus ashei, Juniperus
monosperma and Juniperus pinchotii pollen. It is important to account for the temperature effects,
because ambient temperatures range from below -70 to >30 $^{o}$C. In particular, the altitude of 0.5-
2.0 km to which pollen can be easily transported (Noh et al., 2013) may have temperatures close
to or lower than the chilling temperatures for vegetative species (up to 16.5 $^{o}$C) (Melke, 2015).
Moreover, the temperature in the respiratory tract can reach up to of 37 $^{o}$C (the physiological
temperature). In the work presented here, a vapor sorption analyzer (VSA) was employed to
investigate the hygroscopic growth of pollen grains at different temperature (5 or 15, 25, and 37



ºC), a range covering the chilling temperature to the physiological temperature. Water uptake by
pollen were also examined using diffusion reflectance infrared Fourier transform spectroscopy at
room temperature to complement the VSA results. Furthermore, transmission Fourier
transformation infrared spectroscopy was used to characterize functional groups of dry pollen
grains, in an attempt to seek potential links between chemical composition of pollen grains and
their hygroscopic properties.
**2 Experimental sections**

Six pollen species, all from anemophilous plants, were investigated in this work, including

populus tremuloides and populus deltoides (provided by Sigma Aldrich) as well as ragweed, corn,
pecan and paper mulberry (provided by Polysciences, Inc.).
**2.1 Fourier transformation infrared spectroscopy**

The adsorption of water on pollen samples were studied using in-situ diffusion reflectance

infrared Fourier transform spectroscopy (DRIFTS) at room temperature (~25 ºC). This technique
was described in details in our previous work (Ma et al., 2010), and similar setups have also been
used by other groups to investigate the adsorption of water by mineral dust (Joshi et al., 2017;
Ibrahim et al., 2018). Infrared spectra were recorded using a Nicolet 6700 Fourier transformation
infrared spectrometer (FTIR, Thermo Nicolet Instrument Corporation), equipped with an in-situ
diffuse reflection chamber and a high-sensitivity mercury cadmium telluride (MCT) detector
cooled by liquid nitrogen. A pollen sample (about 10 mg for each sample) under investigation was
placed into a ceramic crucible which was located in the in-situ chamber. A dry air flow and a
humidified air flow were first mixed and then delivered into the chamber, and the total flow rate
was set to 200 mL/min (standard condition). Relative humidity (RH) in the chamber could be
adjusted by varying the flow rate ratio of the dry flow to the humidified flow, and was monitored



online using a moisture meter (CENTER 314). Prior to each experiment, the sample was flushed
with dry air for 3 h at 25 $^{\circ}$C, and the reference spectrum was recorded after the pretreatment.
Infrared spectra were collected and analyzed using OMNIC 6.0 software (Nicolet Corp.). All the
spectra reported here were recorded with a wavenumber resolution of 4 cm$^{-1}$, and 100 scans were
averaged to produce a spectrum. Water adsorption was equilibrated for at least 30 min at each RH
to ensure that the equilibrium between water vapor and adsorbed water was reached.
Pollen samples used in this work were also characterized using transmission FTIR
equipped with a deuterated triglycine sulfate detector (DTGS) detector. Pollen grains and KBr
were mixed with a mass ratio of approximately 1:100 and ground in an agate mortar, and the
mixture was then pressed into a clear disc. Transmission FTIR was employed to examine these
discs, and a pure KBr disc was used as the reference. All the spectra, each of which was the average
of 100 scans, were also recorded at a wavenumber resolution of 4 cm$^{-1}$.

**126    2.2 Vapor sorption analyzer**

Hygroscopic growth of pollen grains was further investigated using a vapor sorption
analyzer (Q5000 SA, TA Instruments, New Castle, DE, USA) described in our previous work (Gu
et al., 2017; Jia et al., 2018). In brief, this instrument measured the sample mass as a function of
RH under isothermal conditions. The instrument can be operated in the temperature range of 5-85
$^{\circ}$C with a temperature accuracy of ±0.1 $^{\circ}$C and in the RH range of 0-98 % with an absolute accuracy
of ±1%. The mass measurement had a range of 0-100 mg and a sensitivity of ±0.01 μg. The initial
mass of each sample used in this work was in the range of 0.5-1 mg. For each of the first three
types of pollen species (populus tremuloides, populus deltoides and ragweed pollen), three samples
in total were investigated, and each sample was studied under isothermal conditions at 5, 25 and
37 $^{\circ}$C. For each of the other three types of pollen species (corn, pecan and paper mulberry pollen),



137 experiments were carried out at 15 ℃ instead of 5 ℃, because during one period the instrument

138 could only be cooled down to 15 ℃ due to a technical problem.

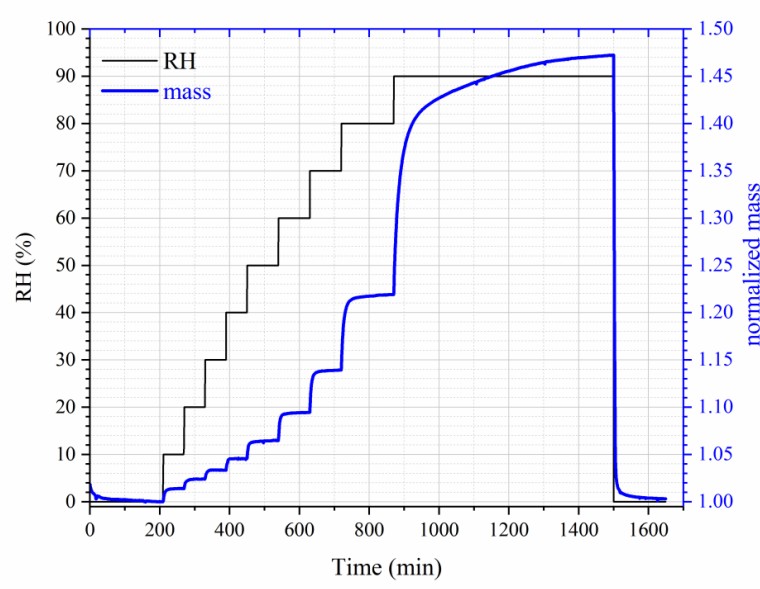


140 **Figure 1.** Change of RH (black curve, left *y*-axis) and normalized sample mass (blue curve, right

141 *y*-axis) with time for a typical experiment in which hygroscopic growth of pollen grains was

142 measured. In this figure a dataset for paper mulberry pollen at 25 ℃ is plotted as an example.


144 For the first sample, at each temperature the sample was first dried at 0% RH (the actual

145 RH was measured to be <1%); after that, RH was increased stepwise to 95% with an increment of

146 5% per step and then switched back to <1% to dry the sample again. At each RH, the sample was

147 equilibrated with the environment (i.e. until the sample mass became stable) before RH was

148 changed to the next value, and the sample mass was considered to be stabilized when the mass

149 change was <0.05% within 30 min. Such a measurement at one temperature could take several

150 days. In order to reduce experimental time, the second and third samples were investigated in a

151 similar way as the first sample, except that RH was increased stepwise to 90% with an increment



of 10% per step. A typical experimental dataset is displayed in Figure 1 as an example to illustrate
the change of RH and normalized sample mass with experimental time.
**3 Results**
**3.1 FTIR characterization of pollen particles**
**3.1.1 Infrared spectra of dry pollen samples**

Figure 2 shows the transmission FTIR spectra of the six pollen species investigated in our

work. A broad band in the range of 3600-3000 cm$^{-1}$, attributed to O-H stretching vibration (Stuart,
2004; Pummer et al., 2013), and two sharp peaks at 2920 and 2850 cm$^{-1}$, attributed to C-H
stretching (Stuart, 2004; Pummer et al., 2013), were observed for all the pollen species. The two
peaks at 1747 and 1658 cm$^{-1}$ were assigned to alkyl ester carbonyls (Pappas et al., 2003; Pummer
et al., 2013), and the two peaks at 1549 and 1458 cm$^{-1}$ (1411 cm$^{-1}$ for paper mulberry pollen) were
assigned to C=C stretching and H-C-H deformation (Stuart, 2004; Pummer et al., 2013). In
addition, the three peaks at 1053, 997 and 845 cm$^{-1}$ were assigned to C-O stretching, C-C stretching,
and C-H out-of-plane bending, respectively (Stuart, 2004; Pummer et al., 2013).





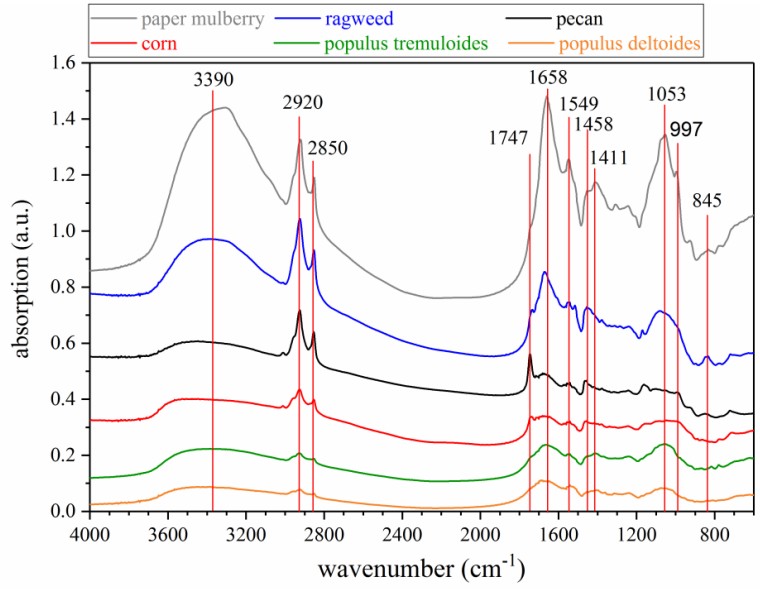

**Figure 2.** Transmission FTIR spectra of six pollen species investigated in this work.

OH groups and C-H groups in organic compounds are generally considered to be hydrophilic and hydrophobic, and one may expect that the amount of OH groups that pollen samples contain may affect their hygroscopicity. In this work we use the intensity ratio of the O-H stretching vibration band (3000-3600 cm$^{-1}$) to the C-H stretching mode (2920 cm$^{-1}$) to qualitatively represent the amount of OH groups pollen samples contain. As shown in Figure 2, the six pollen species examined in our work can be roughly classified into two catalogues: 1) for populus deltoides, populus tremuloides and paper mulberry pollen, the O-H stretching vibration band is more intensive than the C-H stretching mode, indicating that they contain high levels of OH groups; 2) for ragweed, pecan and corn pollen, the O-H stretching vibration band is less intensive than the C-H stretching mode, indicating that they contain low levels of OH groups. The relation between the amount of OH groups that pollen species contain and their hygroscopicity will be further discussed in Section 3.3.





### 3.1.2 Infrared spectra of pollen samples at different RH

In-situ DRIFTS was employed to explore the adsorption of water by pollen grains. Typical spectra of populous deltoides pollen as a function of RH up to 87%, relative to that at <1% RH, are displayed in Figure 3. DRIFTS spectra of other pollen samples at different RH can be found in Figures S1-S5 in the supplement, and are very similar to those for populous deltoides pollen. As evident from Figure 3, several IR peaks (e.g., 3593, 3205, 2135, and 1616 $cm^{-1}$) appeared in the spectra at elevated RH, when compared with that at <1% RH, and their intensities increased with increasing RH. The peaks at 3205, 2135 and 1616 $cm^{-1}$ can be assigned to the stretching, association and bending modes of adsorbed water (Goodman et al., 2001; Schuttlefield et al., 2007; Ma et al., 2010; Hatch et al., 2011; Song and Boily, 2013; Yeşilbaş and Boily, 2016; Joshi et al., 2017; Ibrahim et al., 2018).

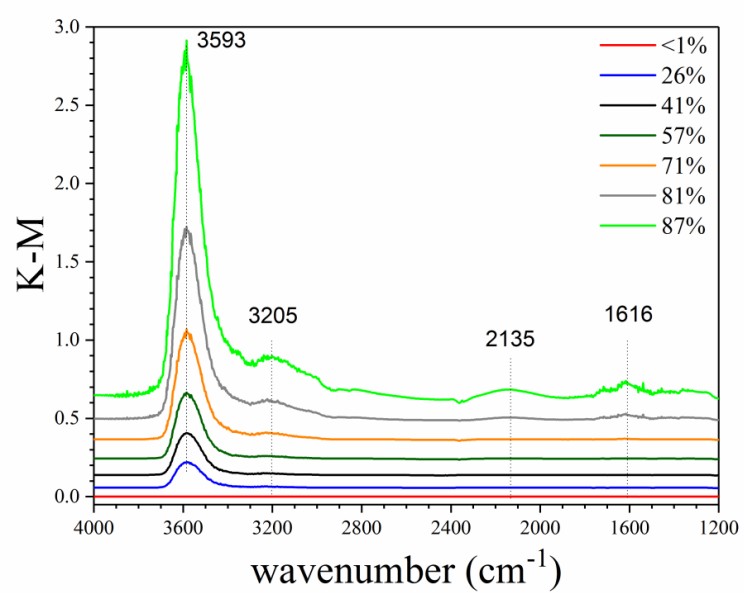

**Figure 3.** In-situ DRIFTS spectra of populous deltoides pollen as a function of RH (<1, 26, 41, 57, 71, 81 and 87%) at 25 °C.



The peak at ~3600 cm$^{-1}$ was the most intensive one observed in the spectra, as shown in
Figure 3. For comparison, the IR peaks assigned to the stretching mode of adsorbed water on
mineral dust and NaCl appeared at lower wavenumbers, typically at around or lower than 3400
cm$^{-1}$ (Schuttlefield et al., 2007; Ma et al., 2010; Tang et al., 2016; Ibrahim et al., 2018). As a result,
the peak at ~3600 cm$^{-1}$ may be assigned to the asymmetric stretching mode of water which
interacted with OH groups in pollen samples (Iwamoto et al., 2003). These results imply that water
adsorption by pollen samples were mainly contributed by OH groups of organic compounds they
contained. The intensities of the IR peaks at ~3600 cm$^{-1}$ were used to represent the amount of
water adsorbed by pollen samples. Table 1 summarizes integrated areas of IR peaks at 3600 cm$^{-1}$
as a function of RH for the six pollen species examined in our work, suggesting that the amount
of adsorbed water by pollen samples increased with RH.





**Table 1.** Integrated areas of IR peaks (at ~3600 cm$^{-1}$) of adsorbed water as a function of RH for the six pollen species investigated in this work. Wavenumber ranges used for integration are 3750-3300 cm$^{-1}$ for populus deltoides pollen, 3750-3350 cm$^{-1}$ for populus tremuloides pollen, 3750-3400 cm$^{-1}$ for ragweed pollen, 3750-3500 cm$^{-1}$ for corn pollen, 3750-3450 cm$^{-1}$ for pecan pollen, and 3750-3300 cm$^{-1}$ for paper mulberry pollen.

| RH (%) | area | RH (%) | area | RH (%) | area |
|---|---|---|---|---|---|
| populus deltoides | | populus tremuloides | | ragweed | |
| 0 | 0 | 0 | 0 | 0 | 0 |
| 26 | 22.7 | 24 | 5.5 | 26 | 10.1 |
| 41 | 36.9 | 41 | 16.4 | 42 | 18.9 |
| 57 | 57.4 | 56 | 35.4 | 50 | 24.5 |
| 71 | 93.6 | 70 | 66.5 | 56 | 30.2 |
| 79 | 137.6 | 78 | 91.2 | 69 | 49.7 |
| 81 | 164.7 | 87 | 156.9 | 88 | 104.6 |
| 87 | 293.1 | | | | |
| corn | | pecan | | paper mulberry | |
| 0 | 0 | 0 | 0 | 0 | 0 |
| 26 | 10.0 | 26 | 8.6 | 26 | 10.2 |
| 42 | 21.5 | 43 | 16.9 | 43 | 17.7 |
| 58 | 41.9 | 58 | 29.5 | 51 | 23.1 |
| 73 | 87.5 | 73 | 60.0 | 59 | 29.8 |
| 89 | 222.2 | 89 | 338.9 | 71 | 46.7 |
| | | | | 86 | 105.1 |

## 3.2 Mass hygroscopic growth

### 3.2.1 Theories

The single hygroscopicity parameter, $\kappa$, is widely used to describe the hygroscopicity of aerosol particles under both subsaturation and supersaturation (Petters and Kreidenweis, 2007). When the Kelvin effect is negligible (this is valid for pollen grains which are typically >1 μm), the





dependence of diameter-based growth factor (GF) on RH can be linked to $\kappa$ via Eq. (1) (Petters
and Kreidenweis, 2007; Tang et al., 2016):
$$RH = \frac{GF^3 - 1}{GF^3 - 1 + \kappa} \quad (1)$$
If we further assume that the particle is spherical, Eq. (1) can be transformed to Eq. (2):
$$\frac{1}{RH} = 1 + \frac{\kappa}{GF^3 - 1} = 1 + \frac{\kappa}{\frac{V}{V_0} - 1} = 1 + \kappa \frac{V_0}{V - V_0} = 1 + \kappa \frac{V_0}{V_w} \quad (2)$$
where $V$, $V_0$, and $V_w$ are the volumes of the particle at the given RH, the dry particle, and water
associated with the particle at the given RH. In order for Eq. (2) to be valid, it is also assumed that
at a given RH, $V$ is equal to the sum of $V_0$ and $V_w$. Eq. (2) can be further transformed to Eqs. (3-4):
$$\frac{1}{RH} = 1 + \kappa \frac{\rho_w}{\rho_p} \frac{m_0}{m_w} \quad (3)$$
$$\frac{m_w}{m_0} = \kappa \cdot \frac{\rho_w}{\rho_p} / (\frac{1}{RH} - 1) \quad (4)$$
where $\rho_w$ and $\rho_p$ are the density of water and the dry particle, and $m_0$ and $m_w$ are the mass of the
dry particle and water associated with the particle at the given RH. Since the particle mass, $m$, is
equal to the sum of $m_0$ and $m_w$, Eq. (5) can be derived from Eq. (4):
$$\frac{m}{m_0} = 1 + \kappa \frac{\rho_w}{\rho_p} / (\frac{1}{RH} - 1) \quad (5)$$
Using an electrodynamic balance, Pope and co-workers (Pope, 2010; Griffiths et al., 2012)
measured the hygroscopic growth of eight types of pollen grains, and found that their mass change
with RH can be approximated by Eq. (5). It should be noted that the original equation derived by
Pope and co-workers (Pope, 2010; Griffiths et al., 2012) has a different format from but is
essentially equivalent to Eq. (5).

The Freundlich adsorption isotherm is another widely used equation to describe the change

of sample mass with RH due to water uptake (Atkins, 1998; Skopp, 2009; Hatch et al., 2011; Tang
et al., 2016):





$$\frac{m}{m_0} = 1 + A_f \cdot \sqrt[B_f]{RH} \quad (6)$$

where $A_f$ and $B_f$ are empirical Freundlich constants related to the adsorption capacity and strength.
In addition, the BET (Brunauer-Emmett-Teller) adsorption isotherm is also widely used to describe
the water adsorption by insoluble solid particles (Brunauer et al., 1938; Goodman et al., 2001;
Henson, 2007; Ma et al., 2010; Tang et al., 2016; Joshi et al., 2017). While the BET adsorption
isotherm typically works well for water adsorption of a few monolayers, the mass of adsorbed
water, as shown in Section 3.2.2, can reach up to 50% of the dry pollen mass at high RH; therefore,
in this work we did not attempt to use the BET adsorption isotherm to describe water adsorption
by pollen grains. Another reason that we did not attempt to use the BET adsorption isotherm is
that the BET adsorption isotherm is mathematically more complex and requires the BET surface
area to be known.
**3.2.2 Mass hygroscopic growth at room temperature**

Figure 4 displays the sample mass (normalized to that at 0% RH) as a function of RH for

pecan pollen at 25 °C. Significant increase in sample mass was observed at elevated RH due to
uptake of water. Compared to that at 0% RH, the sample mass increased by (2.3±0.3)% at 30%
RH, (6.4±0.2)% at 60% RH, (30.3±0.4)% at 90% RH, and up to ~72% at 95% RH. As shown by
the data compiled in Tables S1-S3 in the supplement, substantial increases in sample mass were
also observed for the other five types of pollen species at 25 °C (as well as 5 and 37 °C).





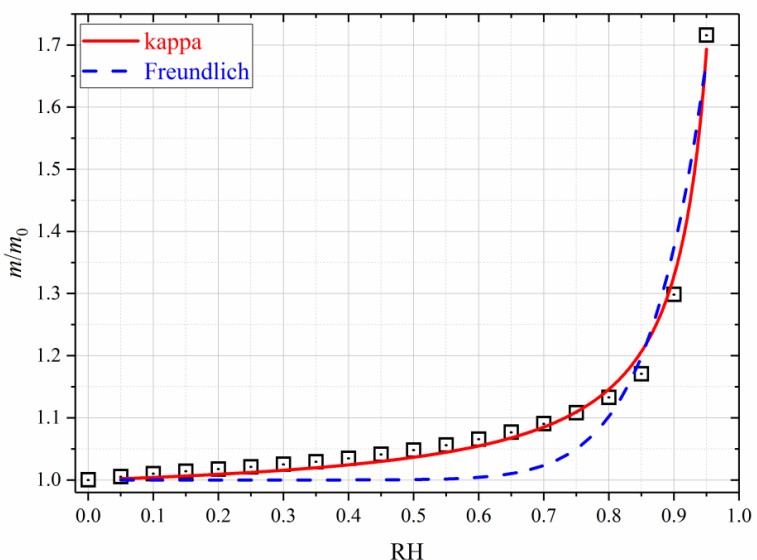


**Figure 4.** Measured change of sample mass (normalized to that at dry conditions, i.e. $m/m_0$) of

pecan pollen as a function of RH (0-0.95) at 25 °C. The experimental data are fitted with the

modified $\kappa$-Köhler theory (solid red curve) and the Freundlich adsorption isotherm (dashed blue

curve).

Hygroscopic properties exhibited considerable variations among different pollen species.

Figure 5a compares the measured ratios of sample mass at 90% RH to that at 0% RH, $m(90\%)/m_0$,

for the six pollen species investigated in this work. We specifically discuss mass changes of pollen

grains at 90% RH (relative to that at 0% RH) because aerosol hygroscopic growth at 90% RH was

widely reported by laboratory and field studies (Kreidenweis and Asa-Awuku, 2014). As shown

in Figure 5a, $m(90\%)/m_0$ determined at 25 °C ranged from 1.293±0.028 (ragweed pollen) to

1.476±0.094 (populus deltoides pollen), i.e. the amount of water adsorbed/absorbed by the six

different pollen species at 90% RH varied between ~30% to ~50% of the dry mass.



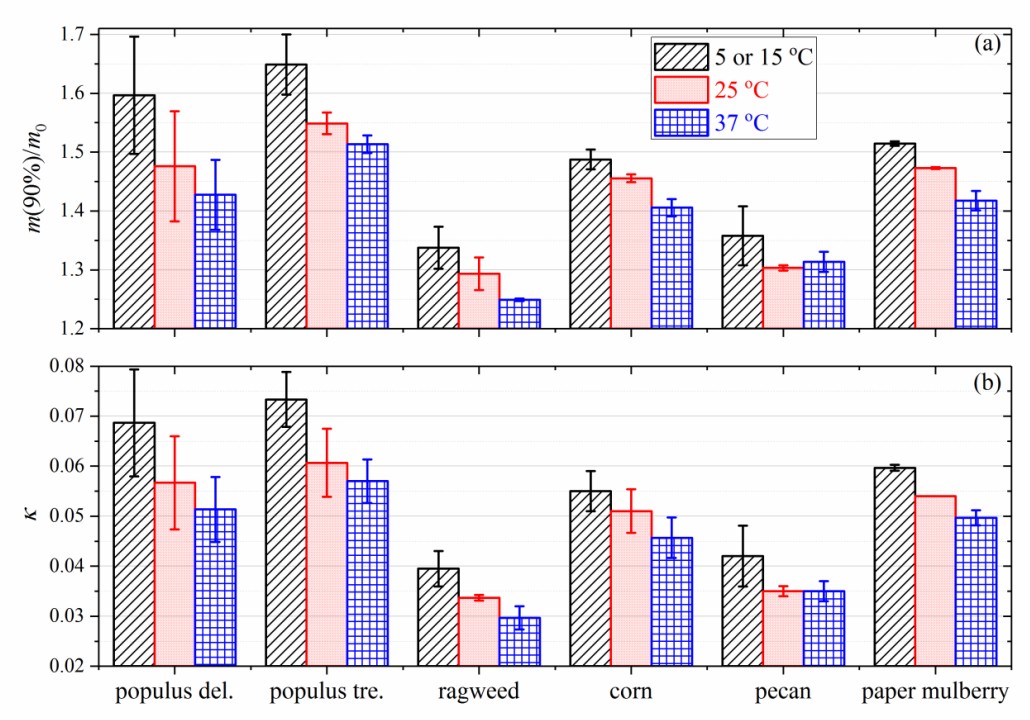

**Figure 5.** Measured ratios of sample mass at 90% RH to that at 0% RH (a) and derived $\kappa$ values

(b) for six pollen species at different temperatures. The lowest temperatures were 5 $^{\circ}$C for populus

deltoides (populus del.), populus tremuloides (populus tre.) and ragweed pollen, and 15 $^{\circ}$C for corn,

pecan and paper mulberry pollen.

As shown in Figure 4, the increase of pecan pollen mass with RH at 25 $^{\circ}$C could be

satisfactorily described by the modified $\kappa$-Köhler theory for the entire RH range (up to 95%). On

the contrary, the Freundlich adsorption isotherm significantly underestimated the sample mass at

low RH, although it represented the experimental data at high RH reasonably well. In addition, we

found that the modified $\kappa$-Köhler theory could also approximate the dependence of sample mass

on RH for all the six types of pollen investigated in this work at different temperatures. If we use





Eq. (5) to fit $m/m_0$ against RH, $\kappa \cdot \rho_\mathrm{w}/\rho_\mathrm{p}$ can be derived. The bulk densities of dry pollen grains were
found to vary with species but typically fall into the range of 0.5-2 g cm$^{-3}$ (Harrington and Metzger,
1963; Hirose and Osada, 2016), and for simplicity $\rho_\mathrm{p}$ was assumed to be 1 g cm$^{-3}$ in this work (i.e.
$\rho_\mathrm{w}/\rho_\mathrm{p}$ is equal to 1). With the assumptions on density and also particle sphericity, $\kappa$ could then be
derived from the measured RH-dependent sample mass at a given temperature.

Table 2 summarizes the average $\kappa$ values at different temperatures for the six pollen species

investigated in this work. At 25 $^\mathrm{o}$C, the $\kappa$ values were found to increase from 0.034±0.001 for
ragweed pollen to 0.061±0.007 for populus tremuloides pollen, varied by almost a factor of 2. The
$\kappa$ values measured by Pope and co-workers (Pope, 2010; Griffiths et al., 2012) were approximately
in the range of 0.05-0.11 (assuming that $\rho_\mathrm{w}/\rho_\mathrm{p}$ is equal to 1), in reasonably good agreement with
these reported in our work.



**Table 2.** Single hygroscopicity parameters ($\kappa$) derived in this work for six pollen species at
different temperatures. All the errors ($\pm 1\ \sigma$) are statistical only.

| pollen type | $T$ (ºC) | sample 1 | sample 2 | sample 3 | average |
|---|---|---|---|---|---|
| populus | 5 | 0.071±0.001 | 0.078±0.001 | 0.057±0.002 | 0.069±0.011 |
| deltoides | 25 | 0.054±0.001 | 0.067±0.002 | 0.049±0.002 | 0.057±0.009 |
| | 37 | 0.058±0.002 | 0.051±0.001 | 0.045±0.002 | 0.051±0.007 |
| populus | 5 | 0.068±0.001 | 0.073±0.001 | 0.079±0.001 | 0.073±0.006 |
| tremuloides | 25 | 0.053±0.002 | 0.063±0.002 | 0.066±0.002 | 0.061±0.007 |
| | 37 | 0.052±0.002 | 0.059±0.002 | 0.060±0.002 | 0.057±0.004 |
| ragweed | 5 | 0.042±0.001 | 0.037±0.002 | -- | 0.040±0.004 |
| | 25 | 0.033±0.002 | 0.034±0.003 | 0.034±0.002 | 0.034±0.001 |
| | 37 | 0.027±0.001 | 0.031±0.002 | 0.031±0.002 | 0.030±0.002 |
| corn | 15 | 0.051±0.001 | 0.059±0.002 | 0.055±0.002 | 0.055±0.004 |
| | 25 | 0.046±0.002 | 0.053±0.002 | 0.054±0.002 | 0.051±0.004 |
| | 37 | 0.041±0.002 | 0.048±0.002 | 0.048±0.002 | 0.046±0.004 |
| pecan | 15 | 0.049±0.001 | 0.038±0.001 | 0.039±0.001 | 0.042±0.006 |
| | 25 | 0.036±0.001 | 0.034±0.001 | 0.035±0.001 | 0.035±0.001 |
| | 37 | 0.033±0.001 | 0.035±0.002 | 0.037±0.001 | 0.035±0.002 |
| paper | 15 | 0.059±0.002 | 0.060±0.002 | 0.060±0.002 | 0.060±0.001 |
| mulberry | 25 | 0.054±0.001 | 0.054±0.001 | 0.054±0.001 | 0.054±0.001 |
| | 37 | 0.048±0.002 | 0.050±0.002 | 0.051±0.002 | 0.050±0.002 |


### 3.3 Discussion

### 3.3.1 Reconciliation between IR and VSA results

Our in-situ DRIFTS measurements, as discussed in Section 3.1.2, suggested that water
uptake by pollen samples was mainly contributed by OH groups of organic compounds they
contained; therefore, it is reasonable to expect that pollen species which contain higher levels of
OH groups would exhibit higher hygroscopicity. Transmission FTIR characterization of pollen
species (Section 3.1.1) showed that populus deltoides, populus tremuloides and paper mulberry





pollen contained high levels of OH groups, and indeed their hygroscopicity ($\kappa$: 0.053-0.054 at 25
$^{o}$C) was higher than the other three pollen species, as shown in Figure 5 and Table 2. For
comparison, ragweed and pecan pollen contained low levels of OH groups and correspondingly
exhibited lower hygroscopicity ($\kappa$: 0.033-0.036 at 25 $^{o}$C). Corn pollen appeared to be an exception:
it contained low levels of OH group but displayed medium hygroscopicity ($\kappa$: ~0.046 at 25 $^{o}$C).
As a result, our results may imply that in addition to chemical composition, other physicochemical
properties, such as porosity and internal structure of pollen grains, could also play an important
role in determining the hygroscopicity of pollen species. One clue came from environmental
scanning electron microscopy observations (Pope, 2010), revealing that pollen grains started to
swell internally before significant water uptake on the surface took place.

In our work two complementary techniques were employed to study the hygroscopic

properties of pollen species. VSA measured the amount of water absorbed/adsorbed by pollen
grains as a function of RH in a quantitative manner, whereas the intensities of IR peaks of adsorbed
water at different RH, as characterized by DRIFTS, can be used semi-quantitatively to represent
the amount of water associated with particles (Ma et al., 2010; Joshi et al., 2017). We compare our
VSA results (i.e. the relative mass change due to water uptake) to the DRIFTS results (i.e.
integrated area of IR peaks at ~3600 cm$^{-1}$). As shown in Figure 6, good correlations between VSA
and DRIFTS results are found for all the six pollen species, suggesting that DRIFTS can be used
to represent the amount of adsorbed water, at least in a semi-quantitative manner.





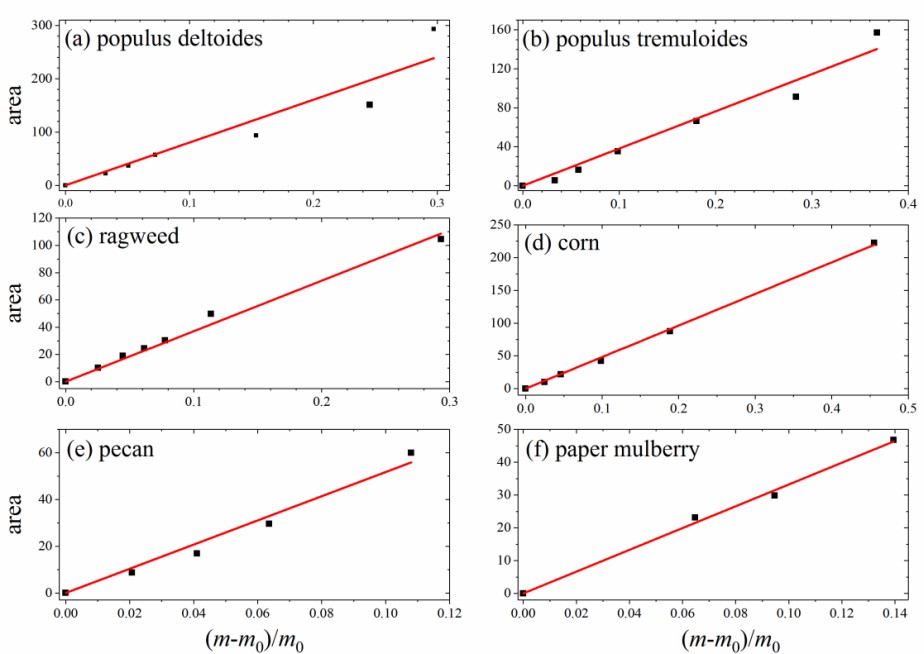


**Figure 6.** Integrated areas of IR peaks at ~3600 cm$^{-1}$ versus relative mass increase due to water

uptake, $(m-m_0)/m_0$, for six pollen species: (a) populus deltoides; (b) populus tremuloides; (c)

ragweed; (d) corn; (e) pecan; (f) paper mulberry.


**3.3.2 Effect of temperature**

Figure 5a shows the comparison of the measured ratios of sample mass at 90% RH to that

at 0% RH, $m(90\%)/m_0$, at different temperatures for the six pollen species. It can be concluded

from Figure 5a that except for pecan pollen for which a small increase in $m(90\%)/m_0$ occurred

when temperature increased from 25 to 37 °C, increase in temperature would lead to small but

nevertheless significant decrease in $m(90\%)/m_0$. For example, $m(90\%)/m_0$ decreased from

1.597±0.100 at 5 °C to 1.476±0.094 at 25 °C and to 1.427±0.060 at 37 °C for populus deltoides





pollen, and from 1.338±0.036 at 5 ºC to 1.293±0.028 at 25 ºC and to 1.249±0.002 at 37 ºC for
ragweed pollen.

We further derived $\kappa$ values at different temperatures for the six pollen species, and the

results are plotted in Figure 5b and summarized in Table 2. Increase in temperature would lead to
decrease in $\kappa$ values, except for pecan pollen. For example, $\kappa$ decreased from 0.073±0.006 at 5 ºC
to 0.057±0.004 at 37 ºC for populus tremuloides pollen, and decreased from 0.060±0.001 at 15 ºC
to 0.050±0.002 at 37 ºC for paper mulberry pollen.
**4 Conclusion and implications**

Pollen grains are one of the most abundant types of primary biological aerosol particles in

the troposphere and play important roles in many aspects of the Earth system. Hygroscopicity is
among the most important physicochemical properties of pollen grains and largely affect their
environmental, health and climatic impacts. However, our knowledge in their hygroscopicity is
still quite limited, and especially the temperature effect has been rarely explored.

In this work we investigated the hygroscopic properties of six types of pollen grains as a

function of RH (up to 95%) at 5 (or 15), 25 and 37 ºC. Substantial increase in pollen mass was
observed at elevated RH due to water uptake for all the six pollen species. Therefore, change in
the mass of pollen grains and their aerodynamic properties at different RH should be taken into
account to better understand their transport and deposition in the troposphere. It was found that the
mass hygroscopic growth of pollen grains can be well approximated by the modified $\kappa$-Köhler
theory. The derived $\kappa$ values at 25 ºC ranged from 0.034±0.001 to 0.061±0.007, varying with
pollen species. DRIFTS measurements indicated that water adsorption by pollen species were
mainly contributed by OH groups of organic compounds contained by pollen grains, and indeed
pollen species that contained lower levels of OH groups (relative to C-H groups, as determined by





transmission FTIR) showed lower hygroscopicity. One exception was corn pollen which contained
low levels of OH group but exhibited medium hygroscopicity, suggesting that in addition to
chemical composition, other physicochemical properties, such as porosity and internal structure,
might play an important role in determining the hygroscopicity of pollen grains. Due to their
moderate hygroscopicity as well as large sizes, pollen grains can thus act as efficient giant CCN
which may have significant impacts on cloud and precipitation (Johnson, 1982; Feingold et al.,
1999; Yin et al., 2000; Posselt and Lohmann, 2008). It is worth noting that only six different pollen
species were examined in our work, and hygroscopic properties of other pollen species commonly
found in the troposphere should be further investigated.

The effect of temperature on the hygroscopicity of pollen grains was systematically

investigated in this work. Increase in temperature (from 5 or 15 $^{\circ}$C to 25 and 37 $^{\circ}$C), a range
covering chilling temperature to physiological temperature, led to small but detectable decrease in
pollen hygroscopicity. For example, $\kappa$ values were found to decrease from 0.073±0.006 at 5 $^{\circ}$C to
0.061±0.007 at 25 $^{\circ}$C and to 0.057±0.004 at 37 $^{\circ}$C for populus tremuloides pollen, and decrease
from 0.060±0.001 at 15 $^{\circ}$C to 0.054±0.001 at 25 $^{\circ}$C to 0.050±0.002 at 37 $^{\circ}$C for paper mulberry
pollen. Our measurements at 37 $^{\circ}$C (physiological temperature) provide very valuable parameters,
which can be used in numerical models to better understand the transport and deposition of pollen
particles in the respiratory system and thus their impacts on human health (Yeh et al., 1996; Broday
and Georgopoulos, 2001; Park and Wexler, 2008; Lambert et al., 2011; Longest and Holbrook,
2012; Tong et al., 2014). Nevertheless, it should be noted that due to the short residence time in
the respiratory system, pollen grains and other inhaled particles in general, may not reach
equilibrium with water vapor in the respiratory tract.



Due to technical challenges, the lowest temperature we could reach in this work was 5 °C,
in the range of normal chilling temperatures for vegetative species and also in the expected
temperature range at the altitudes of 0.5-2.0 km to which pollen grains can be easily transported.
Temperatures in the upper troposphere can be as low as below -70 °C, and it is yet to be explored
whether further decrease in temperature to far below 0 °C will lead to large increase in pollen
hygroscopicity. As a result, experimental measurements of pollen hygroscopicity at lower
temperatures are warranted and would significantly help better understand the transport of pollen
grains in the troposphere. Since water vapor has to be adsorbed or condensed on ice nucleating
particles before heterogeneous ice nucleation can take place (Laaksonen et al., 2016), knowledge
in hygroscopicity and water uptake at temperatures below 0 °C would provide fundamental insights
into atmospheric ice nucleation, in which pollen grains may play an important role (Pratt et al.,
2009; Prenni et al., 2009; Hoose et al., 2010; Pöschl et al., 2010; Murray et al., 2012; Creamean et
al., 2013; Tang et al., 2018).

**Author contribution**

MT, QM and YJL designed the research; WG, CZ, SL and XY did the measurements; MT,
QM, YJL and RJH analyzed the results; MT, QM, YJL and RJH wrote the manuscript with
contribution from all the co-authors.

**Acknowledgment**

This work was funded by National Natural Science Foundation of China (91644106,
91744204 and 91644219), Chinese Academy of Sciences (132744KYSB20160036), Science and
Technology Development Fund of Macau (016/2017/A1), and State Key Laboratory of Organic
Geochemistry (SKLOGA201603A). Mingjin Tang would like to thank the CAS Pioneer Hundred
Talents program for providing a starting grant.





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
