# Peer review of "Water adsorption and hygroscopic growth of six anemophilous pollen species: the effect of temperature"

_Atmospheric Chemistry and Physics, 2018_

## Referee Comment (RC1) · Anonymous Referee #1 · 14 Jan 2019

Tang et al. investigated the hygroscopic growth of six pollen species and its temperature dependence. This study measured water uptake and growth factor by pollen grains using a vapor sorption analyzer and characterize pollen grains using FTIR. A hygroscopic parameter (k) was calculated from the measurements. The subject of this manuscript is within the scope of this journal. There are some minor issues that the authors may want to address before it can be accepted for publication.

1, P5, L98-100, are these pollen species atmospheric relevant? Justification of using these pollen species needs further discussion.

2, P6, L114, what is the uncertainty of this moisture meter?

[Figure]

3, P6, L130-132, which kind of temperature and humidity sensors that can achieve such high accuracy (+-0.1 K and 1% RH) at this temperature and RH range?

4, P7, L137-138, Although it may be fine, I do not think this is an excuse that left the other temperature out. One can simply conduct a few more experiments for the missing points.

5, P8, L157-165, it is suggested to list these peak assignments in a table.

6, P9, L169-173, please justify the use of such ratio as a qualitative representation of OH groups. Have any other studies been using such proxy?

7, P11, L201-203, It only indicates that there is a correlation between water adsorption and OH groups in pollen samples. As discussed in L302-316, there may be other factors contribute to the water uptake. It is suggested to revise these related statements.

8, P12, L215, k parameter is just a fitting from the data. As for now there is no physical meaning for such equation. It is not really a theory.

9, P16, L279-L284, As mentioned above, the k value is obtained from the fitting of these data points, of course, this should fit it well, otherwise k value is wrong. As for Freundlich approach, what are the A and B values? To compare these two different approaches, further discuss is needed.

10, P17, L286-289, If use density of 1 g/cm3, that means k values may be 2 times higher or lower when considering range of 0.5 -2 g/cm3. This is a huge uncertainty. That mean you cannot really compare k values for different species unless they have very similar density.

11, P18, L298, It is not clear what does "All the errors are statistical only." mean.

---

## Referee Comment (RC2) · Anonymous Referee #2 · 22 Jan 2019

The hygroscopicity of pollen species is not well-recognized. The authors investigated six different type of pollen particles using two methods. This work provides valuable dataset for hygroscopicity study community. I have two major comments, which should be addressed and implemented in the revised manuscript. Afterwards, I would like to review another round. (1) In 3.2.1 Theories, the authors assumed the pollen grains are spherical, then, build the link between kappa and mass hygroscopic growth. While, the pollen gains may not the case and are porous in real world. Assuming a spherical particle could lead to a big bias, for example, higher increase in mass, but, smaller hygroscopic growth in diameter. Actually, the mass growth is significant, but the kappa is very small value compared the atmospheric secondary organic aerosols. The authors

only mentioned in line 362-364 that porosity and internal structure, might play an important role in determining the hygroscopicity of pollen grains. But no any discussion in theory part. A detail discussion on the non-spherical situation and its effects on the relationship between kappa and mass growth should be given. (2) For the kappa theory proposed by Petters, 2007, the particles being studied should be assume as solution. Differently, Freundlich adsorption isotherm is water adsorption by materials. The principles between two theories are quite different. The authors may clarify the purpose by using two different theories to fit the observed curve. Which method is more suitable to explain the water uptake of pollen?

---

## Author Comment (AC1) · 29 Jan 2019

Comments by referees are in blue.

Our replies are in black.

Changes to the manuscript are highlighted in red both in here and in the revised manuscript.

**Reply to Ref #1**

Tang et al. investigated the hygroscopic growth of six pollen species and its temperature dependence. This study measured water uptake and growth factor by pollen grains using a vapor sorption analyzer and characterize pollen grains using FTIR. A hygroscopic parameter (k) was calculated from the measurements. The subject of this manuscript is within the scope of this journal. There are some minor issues that the authors may want to address before it can be accepted for publication.

**Reply:** We would like to thank Ref #1 for his/her insightful and detailed comments, which have largely helped us improve our manuscript. We have addressed all the comments adequately in the revised manuscript, as detailed below.

1. P5, L98-100, are these pollen species atmospheric relevant? Justification of using these pollen species needs further discussion.

**Reply:** These pollen species have been chosen in our work primarily because of their commercial availability; nevertheless, these plants are widely distributed in the globe and some of the pollen species, such as ragweed pollen, are well-known due to their impacts on human health. In the revised manuscript (page 5, line 101-105) we have added a few sentences to further justify why these pollen species were chosen in our work: "The six pollen species were chosen in our work primarily because they were commercially available. Furthermore, these plants are also widely distributed in the globe. For example, corn is the most produced grain in the world (International-Grains-Council, 2019), and up to 50% of pollen-related allergic rhinitis cases in North America are caused by ragweed pollen (Taramarcaz et al., 2005)."

2. P6, L114, what is the uncertainty of this moisture meter?

**Reply:** The sensor has an absolute uncertainty of ±2%. In the revised manuscript (page 6, line 118-119) the sentence has been changed to "…and was monitored online using a moisture meter (CENTER 314) with an absolute uncertainty of ±2%."

3. P6, L130-132, which kind of temperature and humidity sensors that can achieve such high accuracy (+-0.1 K and 1% RH) at this temperature and RH range?

**Reply:** The temperature was monitored using a thermocouple, which could achieve a temperature accuracy of ±0.1 K easily. The high accuracy of RH control was achieved by using mass flow controllers to precisely control the flow rates of the dry and humidified nitrogen flows used to regulate RH in the humidity chamber; the accuracy of RH control was routinely checked by measuring the DRH of standard compounds, and the difference in measured and theoretical DRH was always <1%. In the revised manuscript (page 7, line 138-142) we have added a few sentences to explain how high accuracy in RH control was achieved: "RH in the humidity chamber was regulated by using two mass flow controllers to control the dry and humidified nitrogen flows very precisely. The accuracy in RH control was routinely checked by measuring the DRH values for a series of standard compounds, e.g., NaCl, $(NH_4)_2SO_4$, KCl, and etc., and the difference between the measured and theoretical DRH was always <1%."

4. P7, L137-138, Although it may be fine, I do not think this is an excuse that left the other temperature out. One can simply conduct a few more experiments for the missing points.

**Reply:** We agree with the referee, and would like to carry out measurements at 5 $^\circ$C for the other three pollen species. However, unfortunately due to a technical problem, the lowest temperature our instrument could reach was 15 $^\circ$C since the technical problem occurred. In the revised manuscript (page 7, line 146-149) the following change has been implemented for further clarification: "For each of the other three types of pollen species (corn, pecan and paper mulberry pollen), experiments were carried out at 15 $^\circ$C instead of 5 $^\circ$C, because the instrument could only be cooled down to 15 $^\circ$C due to a technical problem after we finished experiments for the first three pollen species."

5. P8, L157-165, it is suggested to list these peak assignments in a table.

**Reply:** As suggested, we have included a table in the revised manuscript (Table 1, page 10) to summarize these peak assignments, and also made corresponding changes to text in Section 3.1.1 (page 8, line 168-169).

6. P9, L169-173, please justify the use of such ratio as a qualitative representation of OH groups. Have any other studies been using such proxy?

**Reply:** As stated in our original manuscript, one may expect that the amount of OH groups (relative to that of C-H groups) that pollen samples contain may affect their hygroscopicity. In addition, a number of previous studies found that heterogeneous aging of organic materials would lead to increase in hygroscopicity; in addition, they also found that the IR absorption intensity for

the O-H stretching mode would increase and the IR absorption intensity for the C-H stretching mode would decrease upon heterogeneous oxidation. Therefore, we used the intensity ratio of the O-H stretching vibration band to the C-H stretching mode to represent the amount of OH groups in a qualitative manner (not quantitatively, however) and explored if there was any correlation between this intensity ratio and measured hygroscopicity. In the revised manuscript (page 9-10, line 181-190) we have expanded our discussion to provide further justification: "OH groups and C-H groups in organic compounds are generally considered to be hydrophilic and hydrophobic, and one may expect that the amount of OH groups (relative to that of C-H groups) that organic samples contain may affect their hygroscopicity. For example, it was found in many previous studies (Eliason et al., 2003; Asad et al., 2004; Hung et al., 2005; Najera et al., 2009) that heterogeneous reactions of organic materials with $O_3$ and OH radicals would increase the IR absorption intensity for the O-H stretching mode and decrease the IR absorption intensity for the C-H stretching mode, meanwhile leading to the enhancement in their hygroscopicity. Therefore, in this work we use the intensity ratio of the O-H stretching vibration band (3000-3600 $cm^{-1}$) to the C-H stretching mode (2920 $cm^{-1}$) to qualitatively represent the amount of OH groups pollen samples contain."

7. P11, L201-203, It only indicates that there is a correlation between water adsorption and OH groups in pollen samples. As discussed in L302-316, there may be other factors contribute to the water uptake. It is suggested to revise these related statements.

**Reply:** As suggested, in the revised manuscript (page 12, line 221-224) we have revised our discussion: "These results imply that water adsorption by pollen samples could be mainly contributed by OH groups of organic compounds they contained; in addition, other factors, such as porosity and internal structure, may also be important for hygroscopic properties of pollen grains."

8. P12, L215, k parameter is just a fitting from the data. As for now there is no physical meaning for such equation. It is not really a theory.

**Reply:** We agree with the referee. In the revised manuscript (page 13, line 236) the title of Section 3.2.1 has been from "Theories" to "Hygroscopicity parameterizations", and throughout the revised manuscript (e.g., page 1, line 34) "$\kappa$-Köhler equation" has been changed to "$\kappa$-Köhler equation".

9, P16, L279-L284, As mentioned above, the k value is obtained from the fitting of these data points, of course, this should fit it well, otherwise k value is wrong. As for Freundlich approach, what are the A and B values? To compare these two different approaches, further discuss is needed.

**Reply:** We respect but do not quite agree with this comment, and would like to clarify it here. In our work we attempted to use both the $\kappa$-Köhler equation and the Freundlich adsorption isotherm to fit our experimental data. For the experimental data shown in Figure 4, if fitted with the $\kappa$-Köhler equation, the best fitting gave a $\kappa \cdot \rho_w / \rho_p$ value of 0.036±0.001, and as shown in Figure 4, the $\kappa$-Köhler equation fitted the experimental data very well; if fitted with the Freundlich adsorption isotherm, the best fitting gave an *A* value of 1.19 and a *B* value of 0.091, and as shown in Figure 4, the Freundlich adsorption isotherm failed to fit the experimental data. This is why we have stated in our original manuscript that the $\kappa$-Köhler equation described our experimental data very well but the Freundlich adsorption isotherm did not.

10, P17, L286-289, If use density of 1 g/cm3, that means k values may be 2 times higher or lower when considering range of 0.5 -2 g/cm3. This is a huge uncertainty. That mean you cannot really compare k values for different species unless they have very similar density.

**Reply:** As pointed out by the referee, our derived $\kappa$ values have large uncertainties, mainly due to large uncertainties in pollen density. However, as shown in Eq. (5), at a given RH mass hygroscopic growth factors, $m/m_0$, depend on $\kappa \cdot \rho_w / \rho_p$ rather than $\kappa$; to compare the hygroscopicity of different pollen species, we should compare $\kappa \cdot \rho_w / \rho_p$ values rather than $\kappa$ values; in our work to make the comparison mathematically simpler, we assume a pollen density of 1 g cm$^{-3}$ and thus $\kappa \cdot \rho_w / \rho_p$ is equal to $\kappa$. As a result, when we compare hygroscopicity of the six pollen species examined in our work, we do not need to assume that they have very similar density.

11, P18, L298, It is not clear what does "All the errors are statistical only." mean.

**Reply:** We actually mean that all the errors given here are standard deviations. In the revised manuscript (page 18, line 333) we have changed this sentence to "All the errors given in this work are standard deviations."

---

## Author Comment (AC2) · 29 Jan 2019

Comments by referees are in blue.

Our replies are in black.

Changes to the manuscript are highlighted in red both in here and in the revised manuscript.

**Reply to Ref #2**

The hygroscopicity of pollen species is not well-recognized. The authors investigated six different type of pollen particles using two methods. This work provides valuable dataset for hygroscopicity study community. I have two major comments, which should be addressed and implemented in the revised manuscript. Afterwards, I would like to review another round.

**Reply:** We would like to thank Ref #2 for his/her insightful and detailed comments, which have largely helped us improve our manuscript. We have addressed all the comments adequately in the revised manuscript, as detailed below.

(1) In 3.2.1 Theories, the authors assumed the pollen grains are spherical, then, build the link between kappa and mass hygroscopic growth. While, the pollen gains may not the case and are porous in real world. Assuming a spherical particle could lead to a big bias, for example, higher increase in mass, but, smaller hygroscopic growth in diameter. Actually, the mass growth is significant, but the kappa is very small value compared the atmospheric secondary organic aerosols. The authors only mentioned in line 362-364 that porosity and internal structure, might play an important role in determining the hygroscopicity of pollen grains. But no any discussion in theory part. A detail discussion on the non-spherical situation and its effects on the relationship between kappa and mass growth should be given.

**Reply:** We agree with the referee, and as suggested, in the revised manuscript (page 17-18, line 323-330) when we discuss $\kappa$ values of pollen species we have added a few sentences to further discuss the particle sphericity assumption and its implications for the derived $\kappa$ value: "It should be noted that in order to convert the measured mass growth to diameter growth and $\kappa$ values, one key assumption is particle sphericity; nevertheless, pollen grains are known to be non-spherical and porous, and therefore our derived $\kappa$ values might be smaller than the actual values. For example, although the mass increase was substantial (around 30-50 % at 90% RH) for the six pollen species examined, their $\kappa$ values at 25 ºC were derived to be in the range of 0.034-0.061, significantly smaller than those (0.1-0.2) for typical secondary organic aerosols produced in smog chamber studies (Petters and Kreidenweis, 2007; Kreidenweis and Asa-Awuku, 2014)."

(2) For the kappa theory proposed by Petters, 2007, the particles being studied should be assume as solution. Differently, Freundlich adsorption isotherm is water adsorption by materials. The principles between two theories are quite different. The authors may clarify the purpose by using two different theories to fit the observed curve. Which method is more suitable to explain the water uptake of pollen?

**Reply:** First of all, as discussed in Section 3.2.2 in the original manuscript, it was concluded in our work that the modified $\kappa$-Köhler equation is more suitable to explain water uptake by pollen because when compared to the Freundlich adsorption isotherm, it fits the experimental data much better.

Furthermore, in Section 3.1.1 of the revised manuscript, we have explained further why we attempted to use these two different equations/theories to fit the experimental data, as detailed below.

We tried to use the modified $\kappa$-Köhler equation because it relates our measured mass growth to the single hygroscopicity parameter. In the revised manuscript (page 14, line 258-262) we have added a few sentences to provide further explanation: "Eq. (5) relates mass growth experimentally measured in our work to the single hygroscopicity parameter ($\kappa$), which has been widely used in atmospheric science to describe hygroscopic properties of aerosol particles under subsaturation as well as their CCN activities under supersaturation; nevertheless, a few assumptions are needed to derive Eq. (5), as discussed."

We also tried to use the Freundlich adsorption isotherm to fit our data because it provides a direct relationship between RH and our measured mass growth, without any additional assumptions. In the revised manuscript (page 14, line 258-262) we have added one sentence to provide further explanation: "One advantage of the Freundlich adsorption isotherm is that it provides a direct relationship between RH and mass growth which was experimentally measured in our work, without any additional assumptions."

---

## Referee Report (RR1)

The second-round review on "Tang et al., Water adsorption and hygroscopic growth of six anemophilous pollen species: the effect of temperature". The questions I concerned in the first-round review were well addressed in the new version. I recommend that this manuscript can be accepted after addressing a major comment below:

The authors may carefully re-consider this conclusion "In-situ DRIFTS measurements suggested that water adsorption by pollen species was mainly contributed by OH groups of organic compounds they contained". The description on "functional groups contribute to water adsorption" may not be seasonable. "OH group" may be "COOH group". Can we say "COOH group" is a major contributor? Or, we can say "OH group" is kind of indicator for hygroscopicity of anemophilous pollen species? Please judge and weigh!

Figure 1: At RH=95%, the normalized mass looks like unstable and keeps increasing. Please make sure the description in the main text (Line 159-160) is identical to that shown in the Figure 1, especially for the mass at RH=95%.

Figure 3 and Table 2, RH<1% is taken in Figure 2, while, RH=0% is used in Table 1. RH=0% is unrealistic.

Figure 4, the RH is presented as 0-0.95, while, in other places, the percentage is used. Please keep the same style throughout the text.

---

## Author Response (AR2)

Atmospheric Chemistry and Physics

February 2019

Dear Professor Jingkun Jiang,

On behalf of all the coauthors, I am very pleased to submit our revised manuscript (**No.**: acp-2018-1118; **title**: Water adsorption and hygroscopic growth of six anemophilous pollen species: the effect of temperature) to **Atmospheric Chemistry and Physics** for consideration of final publication.

We have addressed very carefully the second-round review, which is mostly minor/technical. We have uploaded our reply to the review as well as the revised manuscript with changes highlighted in red.

We would also like to thank you very much in advance for considering our manuscript for final publication.

Sincerely,

Mingjin Tang, PhD

Guangzhou Institute of Geochemistry

Chinese Academy of Sciences

Email: mingjintang@gig.ac.cn

Comments by referees are in blue.

Our replies are in black.

Changes to the manuscript are highlighted in red both here and in the revised manuscript.

The second-round review on "Tang et al., Water adsorption and hygroscopic growth of six anemophilous pollen species: the effect of temperature". The questions I concerned in the first-round review were well addressed in the new version. I recommend that this manuscript can be accepted after addressing a major comment below:

**Reply:** We would like to thank the referee for providing the second-round review. We have addressed all the comments adequately in the revised manuscript, as detailed below.

The authors may carefully re-consider this conclusion "In-situ DRIFTS measurements suggested that water adsorption by pollen species was mainly contributed by OH groups of organic compounds they contained". The description on "functional groups contribute to water adsorption" may not be seasonable. "OH group" may be "COOH group". Can we say "COOH group" is a major contributor? Or, we can say "OH group" is kind of indicator for hygroscopicity of anemophilous pollen species? Please judge and weigh!

**Reply:** We cannot differentiate the relative contribution of C-OH and C(O)-OH groups to water adsorption by pollen samples, and in our work "OH group" is used as a general term to indicate the hygroscopicity of pollen species. In the revised manuscript (page 12, line 224-225) we have added one sentence to further clarify it: "Both C-OH and C(O)-OH groups can contribute to water adsorption by pollen samples, though their relative contribution cannot be resolved in our work."

Figure 1: At RH=95%, the normalized mass looks like unstable and keeps increasing. Please make sure the description in the main text (Line 159-160) is identical to that shown in the Figure 1, especially for the mass at RH=95%.

**Reply:** We have checked the data, and indeed the normalized sample mass became stable at each RH. For the data shown in Figure 1, the time to reach the equilibrium at 95% RH was ~600 min; the mass seemed to be increasing during the entire period (~600 min) but actually became stable in the last 30 min).

Figure 3 and Table 2, RH<1% is taken in Figure 2, while, RH=0% is used in Table 1. RH=0% is unrealistic.

**Reply:** The referee is right. We checked the entire manuscript, and in the revised manuscript all the "0%" have been changed to "<1%" for RH.

Figure 4, the RH is presented as 0-0.95, while, in other places, the percentage is used. Please keep the same style throughout the text.

**Reply:** As suggested by the referee, we have updated this figure in the revised manuscript and the unit used in the revised manuscript for RH is always % (page 15-16).

[revised manuscript text omitted]